# National Versus Local Sustainable Development Plans and Island Priorities in Sanitation: Examples from the Kingdom of Tonga

**Ian White [1,*], Tony Falkland [2] and Taaniela Kula [3]**

1   Fenner School of Environment and Society, Australian National University, Canberra ACT 0200, Australia
2   Island Hydrology Services, 9 Tivey Place, Hughes, Canberra ACT 2605, Australia;
    tony.falkland@netspeed.com.au
3   Natural Resources Division, Ministry of Lands and Natural Resources, Nuku'alofa, Tonga;
    tkula@naturalresources.gov.to
*   Correspondence: ian.white@anu.edu.au; Tel.: +61-418-262-881

**Abstract:** Sanitation, water supply, and their governance remain major challenges in many Pacific Island countries. National sustainable development strategies (NSDSs) are promoted throughout the Pacific as overarching improved governance instruments to identify priorities, plan solutions, and fulfill commitments to sustainable development. Their relevance to local village-level development priorities is uncertain. In this work we compare national priorities for sanitation in NSDSs with those in village community development plans (CDPs) and with metrics in censuses from the Kingdom of Tonga. Tonga's Strategic Development Frameworks (TSDFI 2011–2014 and TSDFII 2015–2025) were developed to focus government and its agencies on national outcomes. From 2007 to 2016, 136 villages throughout Tonga's five Island Divisions (IDs) formulated CDPs involving separately 80% of women, youth, and men in each village. It is shown that censuses in 2006 and 2016 reveal linked improvements in water supply and sanitation systems but identify IDs with continuing challenges. It is found that sanitation and water are a national priority in TSDFI but are absent from the current TSDFII. In contrast, analysis of CDPs, published just after TSDFII, show in one ID, 53% of villages ranked sanitation as a priority and marked differences were found between IDs and between women, youth, and men. CDPs' sanitation priorities in IDs are shown to mostly correspond to sanitation and water metrics in the censuses, but some reflect impacts of natural disasters. Explanations for differences in sanitation priorities between the national and local development plans, as well as suggestions for improving NSDS processes in island countries, are advanced.

**Keywords:** sustainable development strategies; community development plans; small island developing states; governance; sanitation; water supply; hygiene; WASH; census results; top-down versus bottom-up; gender and age; SDG6

## 1. Introduction

The combined challenges of climate change and economic development have long been recognized as major threats to the sustainability of small island developing states (SIDS), impacting especially vital water and sanitation services [1–3]. Water supply and sanitation systems remain continuing challenges in the Pacific region which was the only region in the world where there was an increase in open defecation between 2000 and 2015 [4].

National sustainable development strategies (NSDSs) [5] have been seen as an efficient way for SIDS to identify national challenges, plan their solutions, assign resources and responsibilities,

and fulfill country commitments to international and regional agreements and programs, particularly the United Nations (UN) Sustainable Development Goals (SDGs) and UN SIDS action programs [3]. Part of their efficiency stems from the fact that they are mainly top-down processes, often encouraged, assisted, and supported by external agencies. The relevance of such governance processes imported into dispersed small island communities in Pacific Island countries (PICs) has been questioned [6]. PICs characteristically have well-developed local institutions, resilient social systems, a sensitivity to environmental change, and a high degree of equity [7]. These strengths make PICs well suited to bottom-up planning processes, but these are generally time consuming [8]. In this work, using a valuable example from a multi-island PIC, the Kingdom of Tonga, comparison is made between the planning priorities given to sanitation in top-down national processes and bottom-up village development planning processes.

Improvements in water, sanitation, and hygiene (WASH) have long been recognized as one of the keys for the continued development of PICs [1–3]. The incidence of diarrheal diseases in PICs is, on average, four to five times higher than in larger countries in the Oceania region [9] and is mainly linked to contaminated drinking water as a result of poor sanitation and hygiene [10]. This linkage was one of the reasons that Oceania as a region did not meet the UN's 2015 Millennium Development targets for water and sanitation [11].

Sanitation in SIDS remains a significant challenge for many interacting reasons. Dispersed island communities, restricted land areas, limited and sometimes brackish water supply, El Niño Southern Oscillation (ENSO)- and Pacific Decadal Oscillation (PDO)-related droughts [12], frequent extreme events, including intense tropical cyclones (TCs) [13], easily polluted surface and limited groundwater, coupled with a wide range of population densities, variable island geologies and geomorphologies, and restricted land and financial resources mean that identifying locally acceptable, safe, appropriate, and affordable sanitation systems with minimum impacts on fragile island environments is problematic. The most difficult are islands which rely on variable rainwater as their only source of water for all purposes [10].

Improved governance and accessible information are seen as central to improved WASH outcomes [14], particularly in SIDS. NSDSs have been promulgated as overarching, efficient, integrated governance instruments to identify and solve national development challenges and address UN SDGs [15]. The South Pacific Kingdom of Tonga presents an opportunity to contrast identified sanitation priorities in top-down national and bottom-up village planning processes. Tonga's national development planning instrument, the Tonga Strategic Development Framework 2015–2025 (TSDFII) [16] was launched in 2015 after three months of senior-level consultations and a three-month review by ministries. TSDFII built on experiences from the Tonga Strategic Development Framework 2011–2014 (TSDFI) [17]. TSDFII and TSDFI, present an integrated vision of the direction the government and its agencies' plan to pursue as well as contribute to Tonga's international and regional commitments.

A much lengthier village level development planning process occurred throughout all Island Divisions (IDs) in Tonga between 2007 and 2016. This created Community Development Plans (CDPs) in rural villages throughout Tonga's five IDs under the Ministry of Internal Affairs (MIA) and was facilitated by a non-government organization, Mainstreaming of Rural Development Innovation Tonga Trust (MORDI TT), with support from donors [18]. CDPs ranked local village priority development issues which were prepared and endorsed by a minimum of 80% of each village community. In the past, only men had been involved in local planning. Breaking from tradition, the CDP process also included women and youth who ranked priorities separately from men. In 2016, 136 CDPs out of Tonga's 151 rural villages, were presented to the then Prime Minister [18]. Currently, 117 of the CDPs are available for analysis [19].

It was shown recently [20] that the high priority given to improving water supply in village CDPs universally throughout all Tonga's five IDs contrasts with the omission of water supply from planned outcomes in TSDFII, despite TSDFII's claim that it addresses UN SDG6 on water and sanitation. Here,

the priorities given to sanitation in both TSDFII and in the preceding TSDFI are compared with its ranked priority in the available CDPs. These are also compared with census data from 2006 and 2016 on sanitation type and available water sources at the ID level.

Several indices and indicators have been proposed to monitor performance in WASH [21,22]. The sanitation sustainability index (SSI) has been introduced recently as an indicator for evaluating the sustainability of sanitation systems [23]. The strength of SSI is that it includes integrated measures of the technical, social, environmental, and economic aspects of local community sanitation systems. Unfortunately, in PIC rural villages, even technical information is often absent. What are available usually are national census data on water sources and sanitation types together with often limited hydrological data.

Four questions are addressed in this work:

1. In the absence of detailed information, can census data be used to identify both improvements in sanitation and the diversity of sanitation needs in PICs?
2. Do census data show a link between island sanitation types and island water sources?
3. Do top-down NSDS plans give the same weight to sanitation priorities as those identified in nation-wide bottom-up village development plans and is there an evolution of priorities in NSDSs?
4. Can NSDSs in multi-island countries be improved for sanitation outcomes?

These questions are addressed at the ID level.

Census data on demographics and household water supply sources and sanitation systems from the last two censuses in 2006 and 2016 [24,25] in Tonga are compared at the ID level as well as for the capital area, Greater Nuku'alofa. The relation between sanitation type and water source used are explored. The emphasis given to sanitation in both TSDFI and TSDFII [16,17] is then examined and compared with that given to other infrastructure services. An analysis is presented of the ranked priorities given by women, youth, and men, as well as aggregate village priorities for sanitation in the available CDPs [19] from villages in all IDs. Relationships between CDP priority rankings for sanitation and census sanitation type and water source data are explored. Underlying reasons for differences in top-down versus bottom-up development priorities for sanitation are discussed and options for improving NSDS processes in PICs are given.

## 2. Materials and Methods

### 2.1. Study Location

The Kingdom of Tonga's population of close to 101,000 people live in 169 islands clustered in five IDs with approximate land area of 750 km$^2$. These are dispersed over 700,000 km$^2$ of the southwestern Pacific Ocean (Figure 1). Its islands border the seismically active Tonga trench. Tonga's western islands are volcanic in origin, but its eastern islands are geologically different, being uplifted coral limestone and sand islands. Most of the eastern islands, including Tongatapu, the largest island and location of the capital Nuku'alofa, have a covering of rich volcanic soil deposited from eruptions in the western islands. Volcanic eruptions, earthquakes, tsunamis, TCs, storm surges, and droughts and floods linked to ENSO and the PDO are frequent natural hazards. Recent TCs that have devastated parts of Tonga include TC Ian (2014), TC Gita (2017), and TC Harold (2020).

There is a gradient in annual rainfall across Tonga varying from 1750 mm in the south to 2300 mm in the north, influenced by proximity to the South Pacific Convergence Zone (SPCZ). Annual rainfall variability is moderate with a mean coefficient of variability of 0.21 for the period 1947 to 2019 and decreases from north to south. There are no long-term trends in available annual rainfall [20], which is consistent with climate change projections for Tonga [12]. The Kingdom has a wetter season from November to April followed by a drier season from May to October, also influenced by the position of the SPCZ [26].

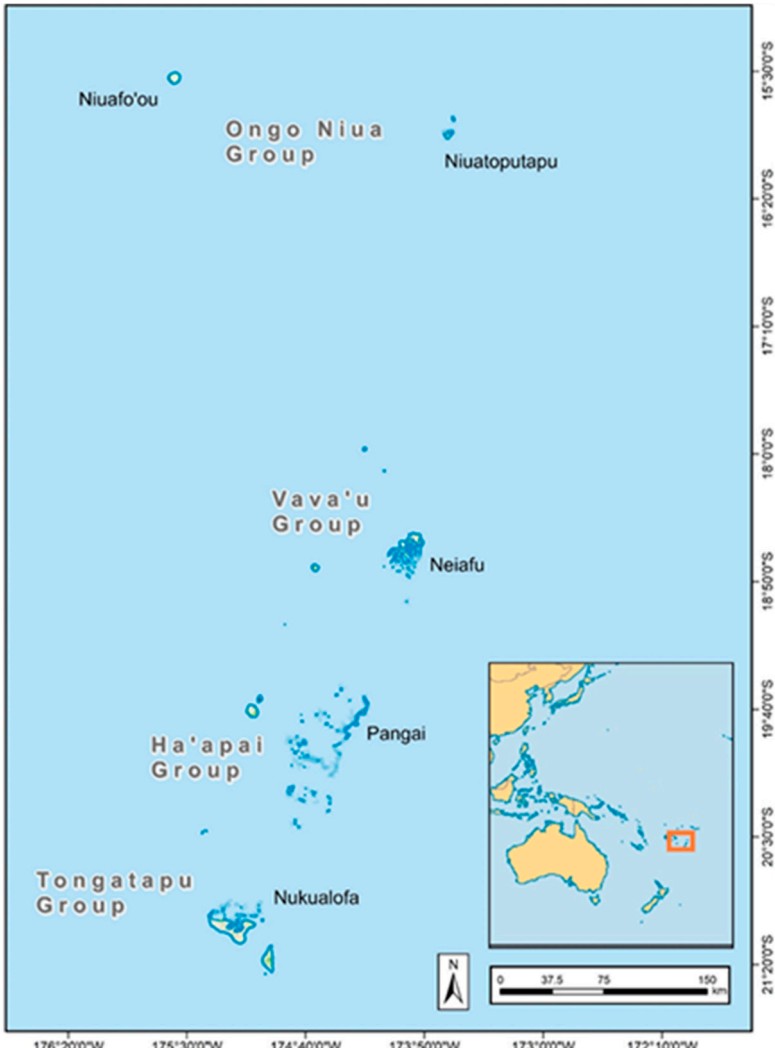

**Figure 1.** Map of the South Pacific Kingdom of Tonga, main Island Divisions and population centers [27].

Water supply in Tonga is mainly sourced from household rainwater harvesting systems and groundwater. Water is supplied from groundwater lenses or springs in the carbonate and sand islands. Many of the volcanic islands do not appear to have useable groundwater [27]. Because seawater underlies groundwater lenses, groundwater salinity in Tonga varies from fresh to brackish and even saline, depending on island geology, size, pumping method and rate, and ENSO and PDO conditions. Springs and groundwater used in 'Eua and most of the groundwater in Tongatapu have lower salinities than groundwater in Vava'u and Ha'apai [27]. Tongans prefer rainwater for drinking, and groundwater is used for all other purposes including toilet flushing. The main sanitation systems are cistern flush toilets or pour flush toilets discharging into septic tanks and pit toilets. Both septic tanks and pit toilets have the potential to contaminate groundwater.

In 2016, about three quarters of Tongans lived on the main island Tongatapu with Greater Nuku'alofa, the capital, having over a third of the country's population. About a quarter of the population is spread over the Kingdom's other four IDs (Figure 1). About 55% of the population is under the age of 25 with youth, aged 14 to 24, making up nearly 19% of the total population [25]. Gross domestic product (GDP) in 2017 was estimated to be US$5900 per person with an annual growth rate around 2.5% [28]. GDP varies widely between IDs with Tongatapu being 15% above the national average in 2013, while Ha'apai was 40% below the national average. Natural disasters have been estimated to cost 4.4% of GDP and TC Ian in 2014, which affected 70% of the population of Ha'apai, severely damaged 75% of the housing stock and cost 11% of GDP [16].



Piped water is supplied from groundwater or spring sources [27]. In Nuku'alofa, 'Eua, and population centers on Vava'u and Ha'apai, piped water is supplied by the Tonga Water Board. Village Water Committees are responsible for piped water supply in villages throughout Tonga. Village piped water supply is mostly intermittent unlike in Greater Nuku'alofa where supply is continuous.

In PICs, traditionally, sanitation was either a personal or household responsibility. In Tonga, sanitation is overseen by the Ministry of Health and, in villages, responsibility is delegated to Village Water Committees (VWCs). There are no public reticulated sewerage systems in Tonga and sanitation is mainly a household responsibility.

### 2.2. Demographics, Water Sources, and Sanitation Systems

The censuses in 2006 and 2016 [16,17] provide details of the national, ID, district, and village level population distribution as well as urban/rural data and can be used to identify changes. This work concentrates on the ID level. Census results are also used here to compare the use of different water sources for all non-drinking purposes, which includes use for sanitation, and to compare sanitation systems used by households across IDs and in Greater Nuku'alofa.

The very limited data on water use from different sources indicates that piped water systems in Tonga supply households (HH) with almost an order of magnitude more water than rain tanks [27]. Here, we make use of the water source ratio (WSR) as a measure of the ID availability of water for all non-drinking purposes [20]:

$$\text{WSR} = (\text{HH supplied by rain tanks})/(\text{HH with piped water supply}) \qquad (1)$$

Sanitation systems in Tonga fall into two groups—those that require little or no water such as pit toilets and pour flush toilets, and cistern flush toilets that require more water. We introduce the sanitation type ratio (STR) as a crude, relative measure of improved sanitation systems:

$$\text{STR} = (\text{HH with pit} + \text{HH with pour flush toilets})/(\text{HH with cistern flush toilets}) \qquad (2)$$

The use of these ratios is a result of the limited data available on water and sanitation systems and their use. We use WSR and STR to compare IDs and the capital Nuku'alofa. If availability of water is a significant factor in the choice of sanitation system [23], then the hypothesis is that the two ratios should be related.

### 2.3. Tonga Strategic Development Framework 2011–2014, TSDFI

TSDF1 [17] was developed after the first full parliamentary elections in Tonga. It built on the consultative process in the previous National Strategic Development Framework in October 2009. Its aim was to provide principles and directions to guide the work of the government administration over its four-year term, 2011 to 2014. It involved consultations over a nine-month period carried out in all five IDs in Tonga [29].

The government's vision for TSDFI was, "To develop and promote a just, equitable, and progressive society in which the people of Tonga enjoy good health, peace, harmony, and prosperity, in meeting their aspirations in life." The government's objectives are expressed at two levels: At the top level are the nine primary Outcome Objectives that contribute to the vision, and at the second level are four enabling themes that support the achievement of these outcomes. The enabling themes are more efficient and effective government, improving macro-economic and fiscal management, sustainable and accountable public enterprises, and coordinated and integrated approach to development partners. The Outcome Objectives have 27 Associated Strategies (ASs) and the enabling themes have a further 15 ASs. The ministries responsible for each Outcome Objective were identified and each were linked to Millennium Development Goals [17].

The Ministry of Finance and National Planning (MFNP) led preparation of TSDFI which was supported by the Australian Agency for International Development (AusAID) and the planning process was facilitated by the Asian Development Bank (ADB).

TDSFI was scanned for references to water, freshwater, rainwater, groundwater, water supply, WASH, sanitation and hygiene, and the Outcome Objectives and ASs were examined for their relevance to sanitation. The relative weight of sanitation to infrastructure and other services was determined.

## 2.4. Tonga Strategic Development Framework 2025–2015, TSDFII

The TSDFII [16] is structured differently to and is more complex than the preceding TSDFI. TSDFII is " ... the overarching framework of the planning system in Tonga. It provides an integrated vision of the direction that Tonga seeks to pursue." [16]. TSDFII is the top of a cascading system of planning and budgeting which is intended to guide:

- Medium term sector and district/island master plans.
- Three year rolling corporate plans and budgets for all ministries, departments, and agencies.
- Annual divisional and staff plans and job descriptions.
- Consultation, monitoring, and evaluation.

TSDFII identifies government priorities, assigns ministerial responsibilities, and aims to focus resources. TSDFII is arranged in a hierarchy where 29 Organizational Outcomes (OOs), grouped under three institutional pillars and two input pillars, feed into seven desired National Outcomes (NOs) that in turn feed into the single Outcome: "More inclusive sustainable growth and development" which supports the single planned National Impact of TSDFII: "A more progressive Tonga supporting a higher quality of life for all" which supports the motto of TSDFII, given by the reformer monarch Tupou I: "God and Tonga are my inheritance". TSDFII also lists 153 Strategic Concepts (SCs) which were issues raised during the consultation process but lie outside TSDFII but are intended as aids to sector, district and ministry, department, and agency planning and budgeting [16].

MFNP led development of TSDFII, supported by the ADB. It used wide but fairly rapid three month high-level consultations with "key sectors of the economy including the community (district and town officers), church leader forums, non-government organizations, and private business forums, including all the main sectors: agriculture, fisheries, tourism, commerce, manufacturing, and construction" [16]. Consultation meetings were held throughout Tongatapu and the IDs of 'Eua, Ha'apai, and Vava'u in the period between October 2014 and December 2014. The northern Ongo Niua ID was not covered. These consultations, together with lessons learnt from TSDFI, were incorporated into drafts of TSDFII which were circulated to government ministries, departments, and agencies for further review in early 2015 [16].

The TSDFII was scanned for references to water, freshwater, rainwater, groundwater, water supply, sanitation, hygiene, and WASH and the planned NOs, OOs, and SCs were examined for their relevance and applicability to sanitation. The weight given to these relative to other infrastructure and other services was determined.

## 2.5. Community Development Plans, 2016

Development of village CDPs [18] began in 2007, under the Local Government Division of the MIA. The CDPs were a response to the then National Vision in TSDFI "A Progressive Tonga Supporting Higher Life for All". Consultations throughout rural villages in Tonga's five IDs were implemented by the NGO Mainstreaming of Rural Development Innovation Trust Tonga (MORDI TT). The CDP process was supported by the International Fund for Agricultural Development (IFAD), UNDP, AusAID, and the Tonga Government. One of the requirements of the project was participation of 80% of the population of each rural village in the development, ranking of priorities, and endorsement of the village CDP. Priorities were ranked separately by women, youth (aged 14 to 24 years), and men. Consultations involved a lengthy process which culminated in District Officers and Town Officers

of 136 village communities presenting their CDPs to the then Prime Minister on 4 October 2016 [18]. The CDPs are really ranked lists of local village concerns.

During the planning process the Department for Local Government was transferred from the Prime Minister's Office to the Ministry of Training, Employment, Youth, and Sports, and then to the MIA [18]. Analysis of and response to CDPs appear to have been deferred by these moves. We have not found any analysis of the valuable information on village development priorities in CDPs apart from our analysis of water supply [20].

Of the 136 CDPs presented, 117 are available on-line [19]. These represent 77.5% of all rural villages in Tonga. CDPs were downloaded and the priority rankings of each village that mentioned sanitation were recorded (Table S1). Particular note was made when sanitation ranked in the top three priorities for women, men, and youth, which also provided a village average. Village level results were aggregated to percentage of villages in each ID. Also recorded was the percentage of villages within each ID that ranked sanitation anywhere in their list of priorities.

## 3. Results

### 3.1. Changes in Demographics

Table 1 shows the demographic data from the 2006 [24] and 2011 [25] censuses for Tonga, for the five IDs, and for Greater Nuku'alofa (GN).

**Table 1.** Demographic data for Tonga, Island Divisions, and Greater Nuku'alofa from the 2006 [24] and 2016 [25] censuses and the percent change that has occurred since 2006.

| Item | Year | Tonga | 'Eua | Tongatapu | Vava'u | Ha'apai | Ongo Niua [1] | Greater [2] Nuku'alofa |
|---|---|---|---|---|---|---|---|---|
| Total Population | 2006 | 101,991 | 5206 | 72,045 | 15,505 | 7570 | 1665 | 34,311 |
| | 2016 | 100,651 | 4945 | 74,611 | 13,738 | 6125 | 1232 | 35,184 |
| *Change (%)* | *2016–2006* | *−1.3* | *−5.0* | *3.6* | *−11* | *−19* | *−26* | *2.5* |
| Population Density (pers/km$^2$) | 2006 | 157 | 60 | 277 | 129 | 69 | 23 | 985 |
| | 2016 | 155 | 57 | 286 | 114 | 56 | 17 | 1010 |
| *Change (%)* | *2016–2006* | *−1.2* | *−5.0* | *3.2* | *−11* | *−19* | *−26* | *2.5* |
| Number of Households (HH) | 2006 | 17,529 | 905 | 12,012 | 2885 | 1377 | 350 | 5753 |
| | 2016 | 18,198 | 889 | 13,096 | 2745 | 1193 | 275 | 6240 |
| *Change (%)* | *2016–2006* | *3.8* | *−1.8* | *9.0* | *−4.9* | *−13* | *−21* | *8.5* |
| Number People per HH | 2006 | 5.8 | 5.8 | 6.0 | 5.4 | 5.5 | 4.8 | 6.0 |
| | 2016 | 5.5 | 5.6 | 5.7 | 5.0 | 5.2 | 4.5 | 5.6 |
| *Change (%)* | *2016–2006* | *−5.5* | *−2.7* | *−5.0* | *−7.0* | *−5* | *−5* | *−6.1* |
| Number of Villages | 2006 | 157 | 14 | 66 | 38 | 27 | 12 | 14 |
| | 2016 | 165 | 15 | 67 | 44 | 27 | 12 | 14 |
| *Change (%)* | *2016–2006* | *5.1* | *7.1* | *1.5* | *16* | *0* | *0* | *0* |
| Area (km$^2$) | | 650 | 87 | 260 [3] | 121 | 109 | 72 | 35 |

[1] Ongo Niua includes Niuafo'ou and Niuatoputapu. [2] Greater Nuku'alofa is made up of the districts of Kolofo'ou and Kolomotu'a in Tongatapu. [3] It appears that the area given for Tongatapu of 260 km$^2$ is the area occupied by villages.

Table 1 shows that in 2016 almost 75% of Tongans live on the main island Tongatapu, with Greater Nuku'alofa having 35% of the total population. About 25% of the population are spread over the Kingdom's other four IDs. The 2016 census defined 77% of the population as rural, although this includes villages within Greater Nuku'alofa [25].

Net migration from Tonga between 2006 and 2016 is evident in Table 1 but with significant inwards migration from the outer IDs to the main island Tongatapu and to Greater Nuku'alofa (Figure 2). The overall population density has decreased correspondingly in Tonga, with significant decreases in population density in Ha'apai and Ongo Niua partially offset by increases in Tongatapu. The population decrease in Ha'apai may be partly due to impacts of TC Ian in 2014 [16].

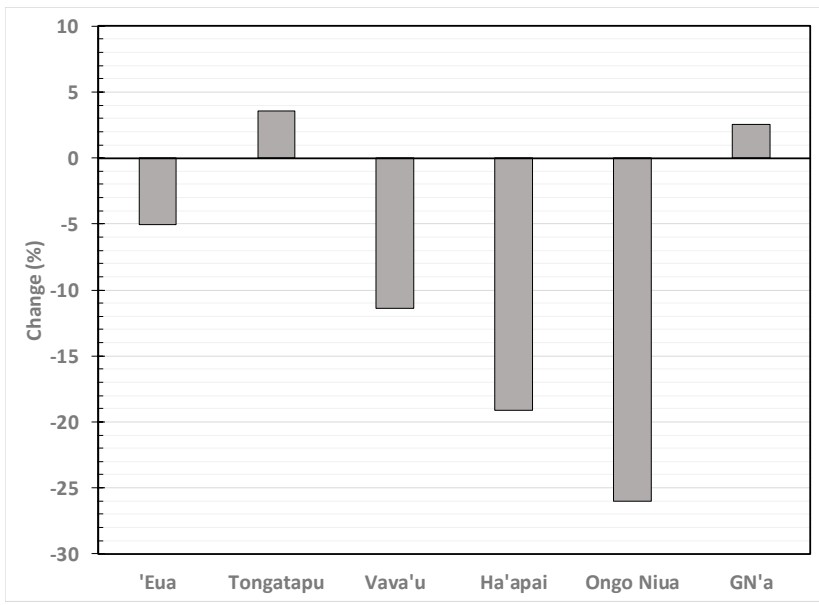

**Figure 2.** Percentage change between 2006 and 2016 in the population of Tonga's five Island Divisions and Greater Nuku'alofa (GN'a) relative to 2006.

Table 1 also reveals an overall increase in the number of households driven by large increases in Tongatapu and Greater Nuku'alofa but offset by decreases in the outer IDs, especially Ha'apai and Ongo Niua. Partly as a result of the shift in population (Figure 2) and the change in the number of households, the number of people per household decreased in Tonga by 5.5% between 2006 and 2016 with similar decreases across all IDs and Greater Nuku'alofa. Paradoxically, there has been an apparent increase in the number of villages in Tonga counted in the censuses, driven mainly by a large apparent increase in Vava'u.

*3.2. Changes in Water Sources*

Because of the intimate connection between water and sanitation, the use of different water sources is first examined. There is limited data on the amount of water used for different purposes in Tonga [20,27,30–32]. It is clear, however, that the greatest volume of household water use is for non-drinking purposes such as toilet flushing, washing, and bathing [33]. Table 2 lists the percentage of households accessing water from different sources for non-drinking purposes and the percentage changes in each between the 2006 and 2016 censuses for Tonga as a whole, the five IDs, and Greater Nuku'alofa.

For Tonga as a whole, Table 2 reveals a 6% increase in access to piped water supply, a 26% decrease in the use of household rank tanks, a large 62% decrease in the use of household water wells, and 40% decrease in other sources (not specified in the censuses) for non-drinking purposes between 2006 and 2016. For the IDs, 'Eua, and Ongo Niua had significant increases in access to piped water but, in Vava'u and Ha'apai, there was very little change in piped water supply. The national decrease in using household rainwater tanks was driven by very large decreases in 'Eua and Tongatapu and a smaller decrease in Ongo Niua. These decreases could be the result of improvements in piped water supply between 2006 and 2016 in 'Eua, Tongatapu, and especially Nuku'alofa, but also in Ongo Niua [27,34]

Although there was limited overall use of household wells in 2006, there were dramatic decreases in their use in 2016 in all IDs. In 'Eua and Vava'u household wells were abandoned as a source of water for non-drinking purposes in the 2016 data and use substantially reduced in Tongatapu, Ha'apai, and Greater Nuku'alofa. Household groundwater wells are easily contaminated by household sanitation systems such as pit toilets and leaking septic tanks [27].

**Table 2.** The percentage of households in Tonga, each Island Division, and Greater Nuku'alofa accessing water for non-drinking purposes from different water sources from the 2006 [24] and 2016 [25] censuses and the percentage change in sources relative to 2006.

| Water Source | Year | Percentage of Households (%) | | | | | | |
|---|---|---|---|---|---|---|---|---|
| | | Tonga | 'Eua | Tongatapu | Vava'u | Ha'apai | Ongo Niua | Greater Nuku'alofa |
| **Piped Supply** | 2006 | 83.2 | 80.1 | 88.5 | 80.8 | 51.7 | 52.1 | 87.5 |
| | 2016 | 88.3 | 95.6 | 93.3 | 80.7 | 52 | 59 | 92 |
| *Change (%)* | *2016–2006* | *6.1* | *19* | *5.4* | *−0.2* | *0.5* | *13* | *5.1* |
| **Rain Tank** | 2006 | 14.7 | 19.1 | 10.0 | 16.8 | 40.1 | 47.3 | 10.4 |
| | 2016 | 10.9 | 4 | 6 | 18.9 | 44.6 | 40.3 | 7.1 |
| *Change (%)* | *2016–2006* | *−26* | *−79* | *−40* | *13* | *11* | *−15* | *−32* |
| **Own Well** | 2006 | 1.6 | 0.3 | 0.9 | 2.2 | 7.8 | 0 | 1.2 |
| | 2016 | 0.6 | 0 | 0.5 | 0 | 3.1 | 0.4 | 0.6 |
| *Change (%)* | *2016–2006* | *−62* | *−100* | *−41* | *−100* | *−60* | *-* | *−52* |
| **Other** | 2006 | 0.5 | 0.4 | 0.6 | 0.2 | 0.4 | 0.6 | 0.9 |
| | 2016 | 0.3 | 0.5 | 0.3 | 0.3 | 0.3 | 0.4 | 0.3 |
| *Change (%)* | *2016–2006* | *−40* | *12* | *−49* | *44* | *−18* | *−30* | *−65* |

Because non-drinking water use is the largest volumetric demand for water [33], the WSR, defined in Equation (1), provides a measure of the difficulty of meeting non-drinking water demand in villages and islands. This is because roof areas used to harvest rainwater are usually small and are often not very efficient, and average household size (Table 1) is relatively large [15]. The available data indicates that in outer islands, rain tanks supply about 20 L/person/day [30–32]. Figure 3 compares the WSP for Tonga's IDs as well as Greater Nuku'alofa for the 2006 and 2016 censuses.

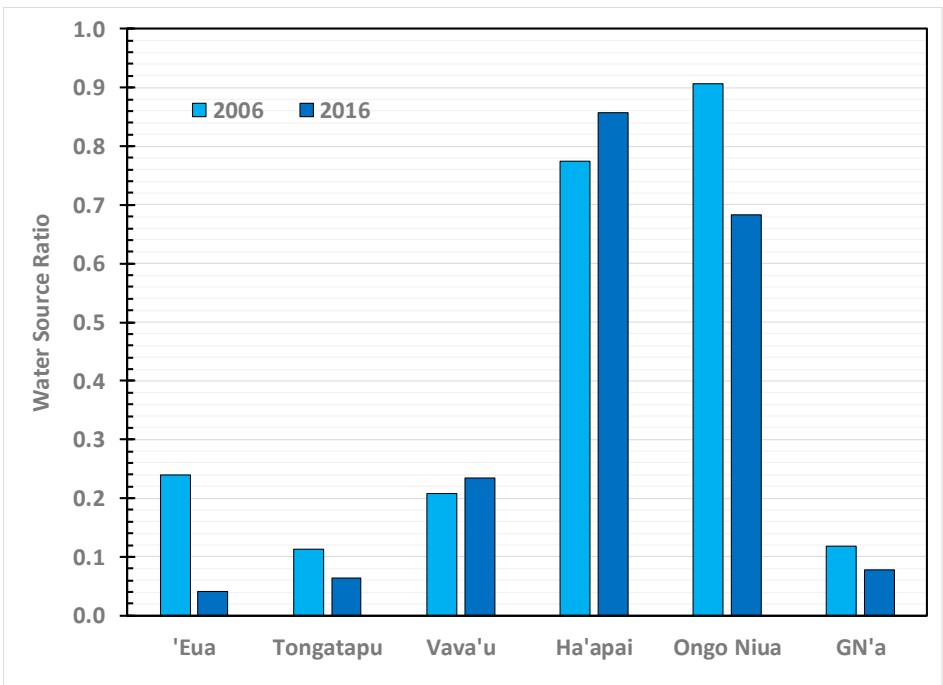

**Figure 3.** Water source ratio, defined in Equation (1), for non-drinking water uses in Tonga's Island Divisions and Greater Nuku'alofa (GN'a) using data from Table 2 for 2006 and 2016 censuses.

Ha'apai and Ongo Niua IDs stand out in Figure 3, with a much larger proportion of their non-drinking water use being supplied from household rain tanks. The increased access to piped

water between 2006 and 2016 in 'Eua, Tongatapu, Ongo Niua, and Greater Nuku'alofa is also evident in Figure 3. In Vava'u and Ha'apai, WSR actually increased between 2006 and 2016 indicating a greater dependency on rain tanks. In these Island Divisions and in Ongo Niua, toilet flushing, washing, and bathing are more dependent on rainwater harvesting than elsewhere in Tonga.

### 3.3. Changes in Sanitation Systems

Table 3 lists the percentage of households using different types of sanitation systems in Tonga, in IDs, and in Greater Nuku'alofa from the 2006 and 2016 censuses.

**Table 3.** The percentage of households in Tonga, Island Divisions, and Greater Nuku'alofa using different types of toilets and the percentage changes between 2006 [24] and 2016 [25] relative to 2006. Flush is a cistern flush toilet; manual is a pour flush toilet and pit is a pit toilet.

| Toilet Type | Year | Percentage of Households (%) | | | | | | |
|---|---|---|---|---|---|---|---|---|
| | | Tonga | 'Eua | Tongatapu | Vava'u | Ha'apai | Ongo Niua | Greater Nuku'alofa |
| **Flush** | 2006 | 70.2 | 60.4 | 80.2 | 53.9 | 38.1 | 33.8 | 87.5 |
| | 2016 | 82.3 | 77.4 | 88.2 | 71.2 | 52.6 | 62.3 | 92.5 |
| *Change (%)* | *2016–2006* | *17* | *28* | *10* | *32* | *38* | *84* | *6* |
| **Manual** | 2006 | 11.4 | 11.0 | 14.2 | 3.4 | 6.0 | 2.6 | 10.6 |
| | 2016 | 8.5 | 7.3 | 9.5 | 4.3 | 7.2 | 8.4 | 6.6 |
| *Change (%)* | *2016–2006* | *−26* | *−33* | *−33* | *25* | *21* | *227* | *−38* |
| **Pit** | 2006 | 18.1 | 27.9 | 5.9 | 42.3 | 55.8 | 63.0 | 1.7 |
| | 2016 | 9.0 | 15.0 | 2.2 | 24.2 | 39.2 | 29.3 | 0.8 |
| *Change (%)* | *2016–2006* | *−51* | *−46* | *−64* | *−43* | *−30* | *−54* | *−53* |
| **Other** | 2006 | 0.02 | 0.00 | 0.02 | 0.07 | 0.00 | 0.00 | 0.03 |
| | 2016 | 0.2 | 0.2 | 0.1 | 0.3 | 1.0 | 0.0 | 0.1 |
| *Change (%)* | *2016–2006* | *918* | *-* | *778* | *376* | *-* | *-* | *180* |

Between 2006 and 2016, dramatic changes are evident in Table 3 in the household use of different toilet systems in Tonga. Nationally, there was a 17% increase in the use of cistern flush toilets with a corresponding significant decrease in the use of manual pour flush toilets and a dramatic drop in the percentage of houses using pit toilets. Although the other toilet category is nationally very small, 1% or less, there was a major increase in use of this unspecified category.

All IDs increased access to household cistern flush toilets over the period 2006 to 2016 with the largest percentage increase in Ongo Niua and the smallest increase in Greater Nuku'alofa due to the already large percentage of households there with cistern flush toilets in 2006. All IDs also showed a large reduction in the use of pit toilets, with the largest reduction being in the main island of Tongatapu and the smallest, although still an appreciable reduction of 30%, in Ha'apai. 'Eua, Tongatapu, and Greater Nuku'alofa showed significant decreases in the use of manual pour flush toilets which, nationally, were offset by increases in their use in Vava'u and Ha'apai and a major increase of almost 230% in Ongo Niua. The "other" toilet category is not specified in either census, so it is uncertain why use of this category increased across all IDs. The 2006 census provided statistics on households which had no access to toilet facilities. This category is missing from the 2016 census and it may be that the increase in the other toilet facilities between 2006 and 2016 in Table 3 is due to the absence of the no toilet category in the 2016 census [24,25].

A measure of improvements in sanitation technology is the STR, defined in Equation (2). Figure 4 plots the STR for Tonga's IDs as well as Greater Nuku'alofa for both censuses.

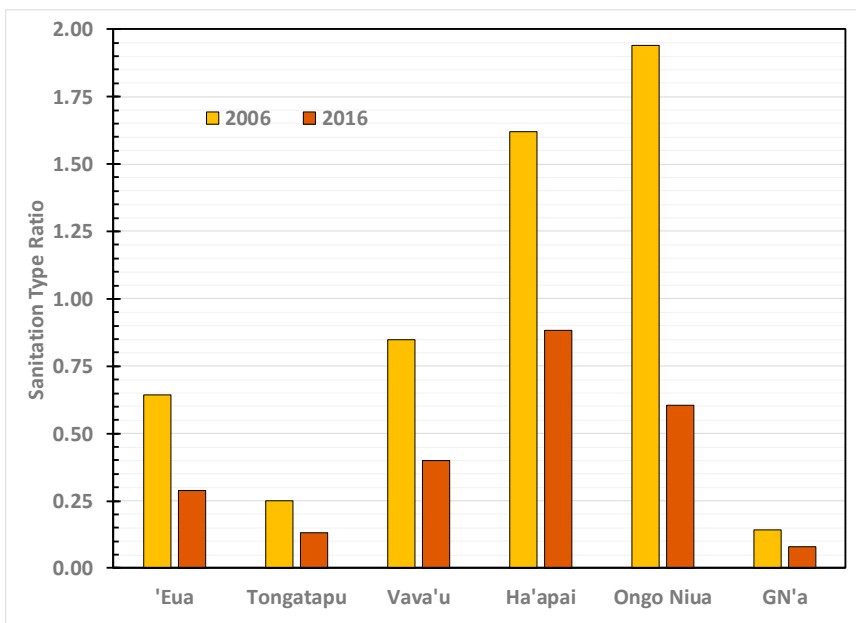

**Figure 4.** Sanitation type ratio, defined in Equation (2), for Tonga's Island Divisions and Greater Nuku'alofa (GN'a) using data from Table 3 for 2006 and 2016.

Figure 4 demonstrates that STR halved between 2006 and 2016 for Tonga and for all IDs and also for Greater Nuku'alofa. In 2006, Ongo Niua had the highest STR with Tongatapu the lowest. STR in Greater Nuku'alofa is half that of Tongatapu in both census years. In 2016, STR is highest in Ha'apai but STR remained lowest in Tongatapu. The results reflect the increases in household use of cistern flush toilets across all IDs in Table 3.

Of the many factors that determine choice of sanitation systems [23], one hypothesis is that the availability of adequate household water supply is a strong determinant. Figure 5 shows the relation between WSR and STR for Tonga's IDs and Greater Nuku'alofa using the 2006 and 2016 census results.

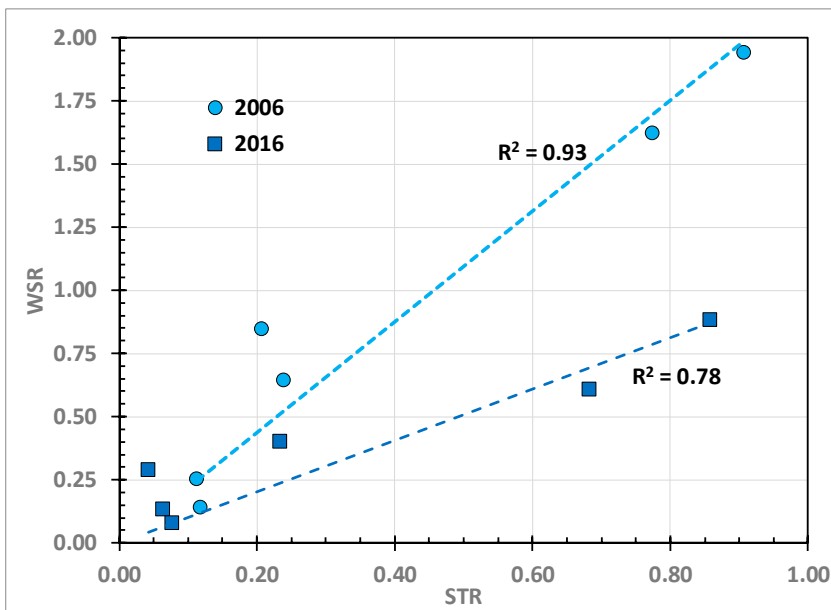

**Figure 5.** Relationship between the sanitation type ratio (STR), defined in Equation (2), and the water source ratio (WSR), defined in Equation (1), for non-drinking water uses for Tonga's Island Divisions and Greater Nuku'alofa (GN'a) from the 2006 and 2016 censuses.

For the 2006 data the relation between STR and WSR is given by:

$$STR = (2.2 \pm 0.1). \ WSR \tag{3}$$

The relation in Equation (3) has coefficient of determination $R^2 = 0.93$ and is highly significant ($p < 5.10^{-6}$). The relation for the 2016 data is:

$$STR = (1.0 \pm 0.1). \ WSR \tag{4}$$

The relation in Equation (4) has $R^2 = 0.78$ and is very significant ($p < 5.10^{-4}$). Equations (3) and (4) reveal that the census data demonstrate a strong correlation between the sanitation type used by households in Tonga and the source of water available to meet non-drinking water demands. They also show the marked improvement in STR relative to WSR between 2006 and 2016.

### 3.4. Sanitation Priorities in TSDFI

To support the Government's vision for Tonga, given in Section 2.3, the TSDFI identified nine Outcome Objectives which are listed in Table 4 together with the number of Associated Strategies (ASs) for each Outcome Objective.

**Table 4.** The Outcome Objectives and Associated Strategies of TSDFI. Shown in parentheses are the number of sub-Associated Strategies [17].

| Outcome Objectives | Number of Associated Strategies |
|---|:---:|
| 1. Strong inclusive communities | 4 |
| 2. Dynamic public and private sector partnership as the engine of growth | 2 (12) |
| 3. Appropriate, well planned, and maintained infrastructure that improves the everyday lives of the people and lowers the cost of business | 8 |
| 4. Sound education standards | 2 |
| 5. Appropriately skilled workforce to meet the available opportunities in Tonga and overseas | 2 |
| 6. Improved health of the people | 2 |
| 7. Cultural awareness, environmental sustainability, disaster risk management, and climate change adaptation, integrated into all planning and implementation of programs | 3 |
| 8. Better governance | 2 |
| 9. Safe, secure, and stable society | 2 |
| **Total** | 27(12) |

Almost 30% of the ASs in Table 4 are linked to the Infrastructure Outcome Objective 3.

The word "sanitation" occurs 10 times throughout TSDFI. In 9 out of the 10 occurrences it occurs in the phrase "water and sanitation", emphasizing the intimate link between them. Table 5 compares the emphasis given to various services linked to infrastructure with that given to water and sanitation. Because of cross references in the ASs, several of the services are mentioned in more than one AS so the total of mentions is more than the number of ASs.

In Table 5, nearly 17% of the total references to services in ASs in TSDFI concern water and sanitation. This includes a mention of solid and liquid wastes under the wastes AS, part of which involves the disposal of septage from septic tanks. The emphasis on water and sanitation is equal to that for energy and for information and communications and half that given to transport.

**Table 5.** The number of references to different services in Associated Strategies under the Infrastructure Outcome 3 in TSDFI.

| Service | References in Associated Strategies |
| --- | --- |
| Energy | 2 |
| Transport | 4 |
| Information and Communications | 2 |
| Wastes | 1 |
| Water | 1 |
| Water and Sanitation | 2 |
| **Total** | 12 |

One of the 27 ASs (nearly 4%) solely concentrates on water and sanitation under Infrastructure Outcome Objective 3. TSDFI lists AS 13: "Maintaining and expanding access to safe water and sanitation for all communities." TSDFI provided performance indicators for this Strategy:

- Proportion of population using an improved drinking water source;
- Proportion of population using an improved sanitation facility;
- Installation of sewerage treatment plant (a septage waste disposal facility on Tongatapu); and
- Proportion of total water resources used.

TSDFI clearly identified water and sanitation as linked national priorities to be included by relevant ministries and agencies in their corporate plans and annual reports. In many PICs, sanitation falls under the Ministry of Health. TSDFI did not, however, include sanitation or hygiene under the Improved Health Outcome Objective 6 (Table 4). Hygiene is only mentioned once in TSDFI in a discussion of the necessity for water and sanitation legislation to ensure "improvement of the hygiene of sanitation facilities". There is no reference to WASH in TSDFI.

*3.5. Sanitation Priorities in TSDFII*

The seven desired NOs in TSDFII [16] are:

A. A more inclusive, sustainable, and dynamic knowledge-based economy.
B. A more inclusive, sustainable, and balanced urban and rural development across Island Divisions.
C. A more inclusive, sustainable, and empowering human development with gender equality.
D. A more inclusive, sustainable, and responsive good governance with law and order.
E. A more inclusive, sustainable, and successful provision and maintenance of infrastructure and technology.
F. A more inclusive, sustainable, and effective land administration, environment management, and resilience to climate and risk.
G. A more inclusive, sustainable, and consistent advancement of our external interests, security, and sovereignty.

The five pillars consist of three institutional pillars and the two input pillars:

**Institutional Pillars:**

1. Economic Institutions
2. Social Institutions
3. Political Institutions

**Input Pillars:**

4. Infrastructure and Technology Inputs
5. Natural Resource and Environment Inputs

These five pillars have 29 linked OOs.

Table 6 shows the number of OOs and SCs focused on services. All are listed under the Infrastructure NO E, except wastes, which was listed under the Environment NO F.

**Table 6.** The number of Organizational Outcomes and Strategic Concepts assigned to infrastructure services in TSDFII [16]. The total number of each are in parentheses.

| Service | Organizational Outcome (29) | Strategic Concepts (153) |
|---|---|---|
| Energy | 1 | 4 |
| Transport | 1 | 9 |
| Information and Communications | 1 | 9 |
| Building and Structures | 1 | 5 |
| Wastes | 1 | 4 |
| Sanitation | 0 | 0 |
| Water Supply | 0 | 1 |

Over 17% of the total OOs in TSDFII are the services listed in Table 6. Energy, transport, information and communications, building and structures, and wastes made up almost 3.5% each of the total OOs listed. Not one OO mentions water and/or sanitation in contrast to TSDFI which identified water and sanitation as a linked national priority under Infrastructure Outcome Objective 3.

Accompanying the OO are 153 SCs. Almost 21% of SCs involved the services in Table 6 but only one references water supply. Transport and information and communications each scored 6% of the SCs in contrast to water, 0.7%, and there is no mention of sanitation in any SC.

Under the natural resources pillar, within OO 5.2, "Improved use of natural resources for long term flow of benefits, SC b) records: "improve the management and delivery of safe water supply for business and households." There are no water supply key performance indicators (KPIs) associated with OO 5.2 SC b), so there is no way to measure performance. Since SCs are only considered an aid to planning there is no pressure for any ministry to address this SC.

Under the Health component of the Social Institutions Pillar, "Percentage of population with safe water supply" is listed as a KPI but has no associated SC or OO. Without this KPI being tied to an OO or an SC there is no guarantee that it would be addressed in ministry corporate plans or reports.

Sanitation is not mentioned in any of the 29 OOs or 153 SCs and it does not appear in any of the KPIs. The word "sanitation" only appears twice in TSDFII. In Annex 1 of TSDFII, "water and sanitation management" is mentioned as one of a long list of action topics of the Small Developing Island Countries Action Agenda in the SAMOA Pathway [3]. In Annex 2, TSDFII lists how NOs contribute to the United Nations SDGs. TSDFII claims that NOs F, E, and B, listed above, all contribute to fulfilling Tonga's commitments to SDG6: "Ensure availability and sustainable management of water and sanitation for all." There are no OOs or SCs under NO F, E, or B involving sanitation and water supply is only included as a single SC under NO F. TSDFII does not include any mention of hygiene or WASH.

The absence of water and sanitation from TSDFII after their inclusion in TSDFI is surprising and leaves the impression that water and sanitation are no longer sustainable development issues in Tonga despite the disparities between IDs revealed in the censuses.

*3.6. Village Community Development Plans (CDPs)*

Village CDPs are designated as plans. In practice, however, the available documents consist of ranked priorities of challenges to village development from women's, youths', and men's perspectives. There are no identified strategies for addressing challenges and no assignment of responsibilities or estimates of resources needed to address priorities. Table 7 provides details of the village CDPs that were available for analysis in each ID [19].

**Table 7.** The number of rural villages in 2016 [26], number of accessible village community development plans for Island Divisions in Tonga, and the medium number of identified priorities identified by women, youth, and men within each Island Division [19].

| Island Division | Number of Rural Villages | Number of CDPs | Median No. of Priorities in Island Division | | |
| --- | --- | --- | --- | --- | --- |
| | | | Women | Youth | Men |
| 'Eua | 12 | 13 | 8 | 7.5 | 7.5 |
| Tongatapu | 53 | 48 | 6 | 5 | 6 |
| Vava'u | 44 | 39 | 7 | 6 | 7 |
| Ha'apai | 27 | 5 | 9 | 7 | 10 |
| Ongo Niua | 12 | 12 | 5 | 5 | 5.5 |
| **Total** | 151 | 117 | | | |
| | **Country Median** | | 7 | 6 | 7 |

In total, 117 CDPs of the original 136 that were presented in 2016 were available on-line [19]. These available CDPs are 77.5% of the 151 rural villages in Tonga (Table 7). CDPs for the main island Tongatapu, excluded the districts of Kolofo'ou and Kolomotu'a that make up Greater Nuku'alofa, so CDPs for Tongatapu in Table 7 represent rural areas of the main island. Table 7 reveals that only five of the 27 villages in Ha'apai had accessible CDPs. This may be due to TC Ian which severely damaged 75% of the housing stock on Ha'apai in 2014 [16].

Table 7 shows for Tonga as a whole, the median number of priority issues identified by women and men were the same while youth identified slightly less. Ongo Niua Island Division villages identified the least number of priority issues while Ha'apai Island Division villages identified the most. Youth in 'Eua identified slightly more priority issues than elsewhere. The maximum number of priorities identified for women, youth, and men in any village was 12, 11, and 12, respectively, while the minimum number was 4, 3, and 3.

### 3.7. Sanitation Priorities in Village CDPs

None of the women, youth, or men in any of the villages throughout Tonga's IDs identified sanitation as their highest priority concern. Only 4% of villages identified sanitation as their second priority with women (8%) nationally having a higher concern than men (3%). None of the youth in any ID rated sanitation as their second priority (0%). The highest average second priority concerns over sanitation were in Ha'apai (7%) and the lowest were in Tongatapu (1%). In Ha'apai women in 20% of villages rated sanitation as their second priority but none of the youth and men in Ha'apai rated sanitation as their second priority (0%).

Table 8 lists the percentage of villages that ranked sanitation within their top three priorities.

**Table 8.** The percentage of villages throughout Tonga's Island Divisions, as well as Tonga collectively, which listed sanitation within their top three priorities in terms of women's, youths', and men's perspectives and the average of all three.

| Island Division | Percentage of Villages with Sanitation within the Top Three Priorities | | | |
| --- | --- | --- | --- | --- |
| | Women | Youth | Men | Average |
| **'Eua** | 8% | 8% | 0% | 5% |
| **Tongatapu** | 2% | 0% | 0% | 1% |
| **Vava'u** | 23% | 3% | 10% | 12% |
| **Ha'apai** | 40% | 0% | 40% | 27% |
| **Ongo Niua** | 0% | 0% | 8% | 3% |
| **Tonga** | **15%** | **2%** | **12%** | **9%** |

Overall, 9% of villages across Tonga in Table 5 placed sanitation within their top three priorities with women again showing more concern about sanitation than men. In contrast, youth in Tonga had

little priority concerns about sanitation. The most concern about sanitation within the top three ranked issues was in the Ha'apai ID with both women and men showing equal concern (40%), in contrast to the absence of concern of youth. Tongatapu showed the least concern over sanitation (1%). Only women in 2% of the villages in Tongatapu ranked sanitation within the top three priorities.

Figure 6 plots the percentage of villages in IDs which identified sanitation as a concern in their list of all identified village priorities.

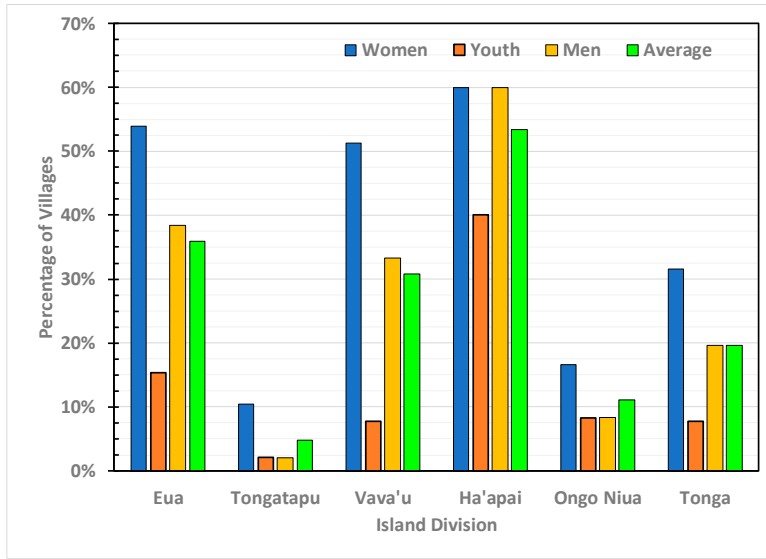

**Figure 6.** Percentage of villages in Island Divisions and in Tonga who ranked sanitation as a priority issue within their list of challenges to village development. Women's, youths', and men's priorities are given together with the average.

For Tonga as a whole, 20% of villages identified sanitation as a priority for village development. Women in Tonga expressed more concern about sanitation (32%) than men (20%) who were more concerned than youth (8%). The ID with the highest overall concerns about sanitation was Ha'apai (53%) with women and men having equal concerns (60%) and youth less (40%). The ID with the lowest number of villages identifying sanitation as a priority was Tongatapu (5%). There, women also expressed more concern (10%) than men or youth (both 2%). Women consistently identified sanitation as a concern in their village, more than men and youth, except in Ha'apai where women and men's concerns were equal.

The CDPs analyzed here were developed between 2007 and 2016, a period covered by both the 2006 and 2016 censuses. To determine if the distribution of the percentage of villages in IDs with percentage of priority sanitation concerns (PSC) is related to the Island distributions of STR (Figure 4), Figure 7 shows the relationship between PSC and STR.

The data in Figure 7 showed no significant relationships between PSC and STR for either census. The results, however, and the impact of a major tsunami on the ID suggest that the data for Ongo Niua may be outliers. If those results are omitted, a strong relationship is found for the other four IDs for the 2006 STR data:

$$\text{PSC}(\%) = (36 \pm 4). \text{ STR} \tag{5}$$

Equation (5) has $R^2 = 0.96$ and is very significant ($p < 0.005$). The relationship for the 2016 STR data is:

$$\text{PSC } (\%) = (68 \pm 10). \text{ STR} \tag{6}$$

Equation (6) has $R^2 = 0.93$ and is significant ($p < 0.01$). The difference in the coefficient between Equations (5) and (6) is due to the decrease in STR between 2006 and 2016 caused mainly by the large decrease in pit toilets.

Because of the strong relationship between STR and WSR, defined in Equations (3) and (4), there are also significant relationships between PSC and WSR for the 2006 Census results:

$$\text{PSC (\%)} = (80 \pm 16).\ \text{WSR} \tag{7}$$

Equation (5) has $R^2 = 0.89$ and is significant ($p < 0.02$). The relationship for the 2016 WSR data is:

$$\text{PSC (\%)} = (69 \pm 24).\ \text{WSR} \tag{8}$$

Equation (6) has $R^2 = 0.74$ and is weakly significant ($p < 0.07$). It is noted that within the standard errors of the coefficients in Equations (7) and (8), the two coefficients are not statistically different ($p > 0.1$). Combining the 2006 and 2016 data for all IDs except Ongo Niua results in the relationship:

$$\text{PSC (\%)} = (73 \pm 9).\ \text{WSR} \tag{9}$$

Equation (9) has $R^2 = 0.90$ and is highly significant ($p < 0.0001$).

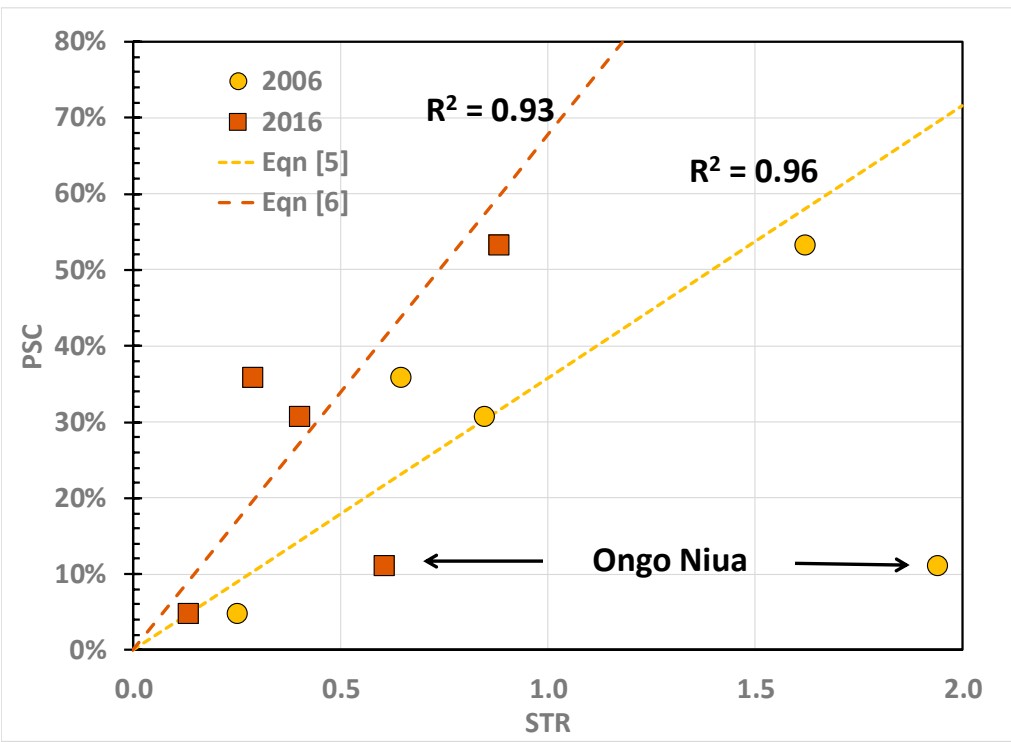

**Figure 7.** The relationship between the percentage of villages in each Island Division with priority sanitation concerns and the sanitation type ratio, defined in Equation (2), from the 2006 and 2016 censuses. The results for Ongo Niua are highlighted.

Equations (5)–(9) suggest that for four of Tonga's IDs, village priority concerns over sanitation are related to the type of sanitation being used which is related to whether water is being sourced from rain tanks or piped water supply, irrespective of which census results are used. The exception is Ongo Niua. It is noted in Figure 7 that Ongo Niua had the largest reduction in STR of any ID between 2006 and 2016, so that sanitation may have been of lesser relative concern there. These results suggest that, with the exception of Ongo Niua, the census results for sanitation type or water source used for non-drinking purposes provide pointers to village priority concerns over sanitation at the ID level. The special case of Ongo Niua will be discussed in the following.

## 4. Discussion

### 4.1. Information from Census Results

Data on water consumed by different uses, particularly sanitation, is sparse in most SIDS, which results in donors relying on average values when planning interventions or emergency relief [33]. Here we have explored the sort of information that can be gleaned about water sanitation from available census results in the specific case of Tonga with its multi-island IDs. The census results (Table 1) show population is concentrated in the main ID Tongatapu, with over a third of the country's population in the capital area, Greater Nuku'alofa. Slightly less than a quarter of the population is scattered across Tonga's other four IDs with population densities lower in northern islands.

Between 2006 and 2016, Tonga's overall population declined slightly but with significant internal shifts in population from the northern and southern-most IDs to Tongatapu and to Greater Nuku'alofa (Figure 2). This increased pressure on water supply and sanitation in the main island. This shift from outer islands to population centers is widespread across PICs [35], and is driven largely by the availability of services, including improved water supply and sanitation services in population centers. In Tonga, natural disasters such as TC Ian [16] and earthquakes and tsunamis [36] also contribute to the population movement.

The census data shows a national shift in water sources for non-drinking purposes between 2006 and 2016. The minimal and declining use of household wells is positive because of the potential for contamination of groundwater by local sanitation systems [10]. There has been increased access to piped water supply and decreased rainwater use, which is valued for drinking, but the shift is uneven (Table 2, Figure 3). In Vava'u and Ha'apai, household access to piped water has hardly changed over the decade but household access to rainwater supply increased in those Divisions. Since this water is used for washing, bathing, and toilet flushing it represents a considerable strain on rainwater harvesting systems, especially during the May to October dry season [10,27]. Even well-managed rainwater systems in Ha'apai are expected to run dry about one year in nine, compared with one year in 28 in Tongatapu and Vava'u and one year in 45 in Ongo Niua [20].

The main types of household toilets in use in Tonga are cistern flush, pour flush, and pit toilets (Table 3). There are no statistics on the use of compost toilets despite an early trial in Ha'apai [37]. Nationally, there has been an increase in household use of cistern flush toilets between 2006 and 2016, a corresponding decrease in pour flush toilets and a large 50% decrease in pit toilets, which is fairly uniform across all IDs and Greater Nuku'alofa. The decrease in pit toilets is particularly important for reducing local groundwater contamination and improving hygiene.

The STR, defined in Equation (2), was introduced as an approximate relative measure of sanitation systems which use less water compared to those requiring more. STR halved in Tonga between 2006 and 2016 with decreases occurring in all IDs, and particularly Ongo Niua. This demonstrates a national shift to sanitation systems with larger water use. Despite this shift, the STR for Ha'apai and Ongo Niua in 2016 remain high compared to other IDs and particularly to Greater Nuku'alofa.

One of the questions this work sought to answer was: "Do census data show a link between island sanitation types and island water sources?" While there are many factors that influence the choice by households of the type of sanitation system [23], particularly in PICs, an adequate supply of water appears crucial. We found very strong and highly significant ($p < 5.10^{-4}$) correlations between STR and WSR for all Tonga's IDs for both 2006 and 2016 census results (Equations (3) and (4)). This implies that, even in Tonga's outer islands, the move to improved sanitation systems is governed by access to adequate water supply. The change in STR revealed in the census data also implies that cistern flush toilets are the desired system by householders. The analysis of census data also showed continuing large differences in reliable water sources and improved sanitation systems available to households between the main island, Tongatapu including the capital region, Greater Nuku'alofa, and the other IDs.

One of the questions this work aimed to address was: "In the absence of detailed information, can census data be used to identify both improvements in sanitation and the diversity of sanitation

needs in PICs?" Comparison of the 2006 and 2016 censuses has shown marked changes in sanitation types used and revealed IDs which continue to be disadvantaged relative to the main island. We will now consider if these changes and differences are reflected in TSDFI and TSDFII.

*4.2. National Versus Local Sustainable Development Plans and Sanitation*

TSDFI, spanning the period 2011 to 2014, recognized explicitly the intimate link between water and sanitation evident in the census results. Of its nine national Outcome Objectives, two are directly relevant to water and sanitation. About 30% of the 27 Associated Strategies in TSDFI concerned infrastructure services, and 17% of them are focused on improving access to safe water and sanitation under the Infrastructure Outcome Objective. There are no water or sanitation Associated Strategies or related performance indicators under the Improved Health Outcome Objective. This is surprising in Tonga. All village water supply systems as well as national sanitation are the responsibility of the Ministry of Health.

TSDFII, covering the period 2015 to 2025, was built on lessons learnt through TSDPI but its structure is more complex and its objectives and strategies more numerous. TSDFII abandoned the linkage between water and sanitation in TSDFI. It contains no mention of water and/or sanitation in any of its 29 Organizational Outcomes. This is despite the fact that TSDFII claims it contributes to Tonga's commitments to SDG6. Of the 153 Strategic Concepts raised in the three-month consultations leading to TSDFII, one concerned water supply but, again, there was no mention of sanitation. This means that ministries have no national obligation to include sanitation in corporate plans or reports. This appears to imply that while water and sanitation were a priority in TSDFI, with the improvements in water supply and sanitation evidenced in the 2006 and 2016 censuses, water supply and sanitation are no longer national priorities in TSDFII, despite the continuing disparity between IDs evident in the censuses.

The available local village CDPs, presented in 2016, were developed over a nine-year period. Ha'apai had the highest median number of priority development concerns, perhaps partly reflecting the impacts of TC Ian in 2014 [16], while Ongo Niua had the lowest (Table 7). Sanitation was not ranked highest priority concern by any village in Tonga, and only 9% of villages ranked sanitation within their top three priorities, with women more concerned than men and both much more concerned than youth. Ha'apai villages had a much higher concern over sanitation than in Tongatapu which had the lowest concerns (Table 8). Nationally, 20% of villages identified sanitation within their list of development priorities (Figure 6). The percentage of villages expressing concern in Ha'apai (53%) about sanitation was an order of magnitude higher that in Tongatapu (5%). Nationally, again women had a higher concern over sanitation as a development priority than men or youth.

The results for sanitation are in contrast to the priority rankings for water supply in the CDPs where 55% of villages identified water supply as highest priority [20]. The lower priority for sanitation appears to stem from the fact that traditionally in PICs, sanitation was either a personal or household responsibility. There is no reticulated sewage system in Tonga. Piped water supply, however, is a community responsibility through Village Water Committees or, in population centers, the Tonga Water Board.

Sanitation priority development concerns expressed by IDs in their CDPs were strongly correlated with the sanitation type and water supply ratios from the census data, Equations (5)–(8), provided the results for Ongo Niua are ignored. This implies that the census results may be useful in identifying sanitation priorities in NSDS processes, certainly at the ID level. The apparent anomalous results in Ongo Niua appear to be due to major improvements in sanitation systems there between 2006 and 2016. Ongo Niua will be further discussed below.

Another of the questions this work sought to answer was: "Do top-down NSDS plans give the same weight to sanitation priorities as that identified in nation-wide bottom-up village development plans and is there an evolution of priorities in NSDS?" It has been shown here that while TSDFI did

recognize water and sanitation as national priorities in line with the village CDPs, TSDFII did not. Suggested reasons for this discrepancy follow.

### 4.3. Why is Ongo Niua an Outlier?

Unlike other IDs in Tonga, the apparent lack of village concerns over sanitation in the CDPs of the remote northern Ongo Niua did not match the water and sanitation metrics in the 2006 and 2016 censuses. Ongo Niua ID is made up of three islands Niuatoputapu, Niuafo'ou, and Tafahi. Niuatoputapu has 57% of the population of Ongo Niua (1232 in 2016) while Tafahi has 2.5%. In Niuafo'ou, household water supplies are from rainwater harvesting [32] and sanitation is mainly provided by pit toilets and pour flush toilets. Niuatoputapu has a piped groundwater supply as well as household rain tanks and households access cistern flush toilets as well.

On 30 September 2009, Niuatoputapu island was struck by an 8.3 magnitude earthquake whose epicenter was 190 km northeast of the island. Three tsunami waves up to 17 m high struck the island, penetrating over one kilometer inland and inundated about 46 percent of the island. This resulted in the death of nine people, the complete destruction of one third of the houses and all government buildings, water supply and sanitation infrastructure [36].

A World Bank post-tsunami reconstruction project [36], completed in 2014, moved houses inland, constructed 73 new houses and a new piped water system, reconstructed roads, improved household rainwater harvesting systems, and installed pedestal cistern flush toilets and septic tanks. The project noted that there were still some pit and compost toilets on the island.

It appears that this reconstruction project is the reason that the STR in Ongo Niua decreased dramatically between 2006 and 2016 (Figure 4) and is probably the reason for the low ranking of sanitation in village CDPs in Ongo Niua. The 2009 tsunami, 2014 TC Ian, 2017 TC Gita, and 2020 TC Harold re-emphasize the vulnerability of sanitation and water supply infrastructure in SIDS to frequent natural disasters.

### 4.4. The Mismatch between National and Local Development Planning for Sanitation

Four possible explanations are advanced for the absence of priority given to water and sanitation in TSDFII in 2015 compared with the emphasis in TSDFI. The first is that the significant national improvements shown in sanitation and water supply in the 2016 census were recognized prior to the census being conducted in 2016 so that water and sanitation were no longer considered a priority for national planning. The second is that, traditionally, sanitation has been a household/individual responsibility, as in all PICs, and with no reticulated sewerage system in Tonga it is not considered government business, even though the government has to dispose of septage. This, however, does not explain that water and sanitation was a linked priority in TSDFI. The third is that all ministries and most of the government agencies with responsibility for TDSFII are located in Tongatapu and mostly in Greater Nuku'alofa where water supply is continuous and relatively plentiful and, as a consequence, improved cistern flush sanitation systems are widely in use. The fourth is that TDSFI was based on nine months of consultations in all IDs whereas TDSFII relied on consultations over three months in four of the five IDs.

### 4.5. Sanitation in Islands Reliant on Rainwater Harvesting

We found IDs with higher dependence on rainwater harvesting have generally more village-level concerns about the type of household sanitation system than in islands with greater access to piped water. The exception was Ongo Niua, where, in some islands, their volcanic geology means that groundwater appears absent or is very limited. Restricted rainwater storage, small roof areas, lack of maintenance, six month dry seasons, ENSO- and PDO-related droughts, and significant household demand mean most rainwater harvesting systems are not able to supply reliably sufficient water for flush toilets.

Several solutions have been trialed in small remote island countries and a range of zero or minimal water use systems are available [23]. Pit toilets do not use water and are relatively cheap, but they are regarded as less convenient and do pollute groundwater. Compost toilets were trialed in Lifuka in the Ha'apai ID [37], but they are not included in census surveys. CDPs coupled with census results show a general preference for flush toilets which are more convenient and require less upkeep than compost toilets. The use of seawater or brackish water for flushing has been used in high population density atolls either from reticulated supply systems or local wells. Leakage from saline reticulation systems can salinize fresh groundwater [10]. Desalinated water can also be used to provide water for flushing. However, desalination systems are expensive to operate and difficult to operate and maintain in remote outer islands [10].

While a number of economic, social, cultural, environmental, technical, and cultural factors influence the choice of household sanitation systems [23], we have shown here that an adequate supply of water is a major factor. Remote island communities look to the population centers and see flush toilets as the standard. Indeed, in Tonga, cistern flush toilets discharging into septic tanks are the desired goal. This is despite the fact the normal construction of septic tanks with concrete blocks inevitably leads to leakages with the potential to pollute soil and groundwater. Molded plastic septic tanks appear a safer but more expensive option [10].

### 4.6. Sanitation, a Dilemma for Island Governments

The dramatic change in emphasis on water and sanitation between TSDFI and TSDFII is evidence of a dilemma many PIC governments face. Water supply traditionally was the responsibility of the extended family, while sanitation was largely a personal matter which has evolved to a household responsibility. The transition to modernity in PICs has resulted in high density urban centers where water and sanitation are often government responsibilities but with lower density rural areas where tradition is often still strong. Addressing the diversity of needs in disparate situations equitably and responding to international programs and agreements such as SDG6, which focus on government responsibilities in sanitation, counter long-held traditional thinking.

In Tonga the situation is more complex. Most households prefer rainwater for drinking, even in the capital area, with government agencies or Village Water Committees supplying piped groundwater for other uses, where possible. Households, therefore, still play a role in water supply, which is more prominent in rural communities. Even in the capital and population centers, sanitation is a household responsibility with government responsible for collecting and disposing of septage in some areas. So, there is uncertainty about the appropriate role of government in sanitation as evidenced by its absence as a national priority from TSDFII. Clearly, implementing improved health, building, and environmental standards nationally are key government functions in sanitation and the facilitation of improved sanitation facilities through micro-financing schemes is a possibility [23]. Here we have shown that rural women are more concerned about sanitation than men or youth. Perhaps the election of more women to parliament would sharpen the national focus on sanitation?

### 4.7. Limitations of this Work

This work only examined aggregated ID level data for the census and CDP results. Both contain a wealth of information at the district and village level which could better inform both planning and intervention. We have not used here more sophisticated sanitation or WASH indices [21–23] to identify differences between IDs. This is due to the limited information available at the ID, district, and village level. Even per capita water consumption is not available except for the capital and population centers [27], and a handful of villages [30–32]. The recent passage of Tonga's Water Resources Act 2020 provides stimulus for improved data collection across the country.

The crude measures we have used here, WSR and STR, have inbuilt assumptions. The WSR assumes that all piped water systems in IDs are comparable. They are not. Greater Nuku'alofa and major centers in three other islands have access to a continuous supply of water, but in most rural

villages piped water systems operate intermittently requiring household storage for continuous use. The STR assumes that flush toilets are what island communities desire irrespective of the adequacy of water supply. In the case of Ongo Niua, the large decrease in STR documented reconstruction efforts following a major natural disaster.

## 5. Conclusions

WASH has been identified as a major continuing national challenge in PICs [9] and improved governance is seen as a key step in addressing that challenge [14]. National sustainable development strategies and plans have been advocated as efficient governance instruments to identify overarching national priorities, select appropriate solutions, allocate responsibilities and resources, and fulfil international and regional commitments, especially in sustainable development. Inevitably, they are largely top-down processes. Transplanted governance institutions from one country may not suit another and standardized processes or formulaic approaches may not be apposite in PICs [6,20], particularly those with village communities dispersed over islands spread across large ocean and with a wide range of development challenges and traditional practices.

PICs have ideal strengths for bottom-up processes [7]. One of the inherent drawbacks of bottom-up processes is the inevitable time and cost penalties they incur. The Kingdom of Tonga provided the opportunity to examine how both top-down and bottom-up planning processes handle sanitation and water at the Island Division level. Consultations leading to TSDFII took three months, TSDFI took nine months, while the community consensuses building leading to CDPs evolved over nine years.

The previous two censuses in Tonga provide a clear picture of the household use of sanitation types and water sources, the changes in usage over time, and show large variations between the main island Tongatapu, especially in the capital area, and the other Island Divisions. The census data revealed that the type of sanitation system chosen is strongly correlated with access to piped water supply at the Island Division level. Censuses provide valuable data about differences in sanitation type and water sources which correlated well with aggregated village sanitation priorities in most Island Divisions. These relationships may also be evident in other PICs. The changes in sanitation metrics also revealed responses to natural disasters, so frequent in SIDS.

The first Tonga Strategic Development Framework, TDSFI, from 2011 to 2014 acknowledged the linkage between sanitation and water and their importance under an Infrastructure Outcome Objective. There is no recognition in the current TDSFII of that linkage and no mention of improved water and sanitation services under the Infrastructure Pillar or within the Health Outcome Objectives. The available bottom-up village CDPs show a markedly different set of priorities for water [20] and sanitation across Island Divisions, with the main Island Division being largely satisfied with the sanitation available while outer islands, and particularly Ha'apai, had significant development concerns over sanitation. One of the strengths of the CDP process was that it provided women and youth with the opportunity to voice their concerns as well as men who are normally the only group consulted. Overall, village CDPs showed higher concerns over sanitation by women than men with youth the least concerned.

A question raised here is: Can NSDS in multi-island countries be improved for sanitation and water outcomes? We have found here a major difference in the national priority given to sanitation and water in TSDFI and the diverse sanitation priorities expressed in village CDPs, especially by women, and their total absence in TSDFII. TSDFI reflects the sanitation data in the 2006 census. TSDFII does not acknowledge the large differences in sanitation systems evident in either the 2006 or 2016 censuses. Instead, TSDFII appears to reflect the lack of village priority concerns over sanitation found in village CDPs from the main island, Tongatapu.

Refining census questions to better target sanitation types and sources and reliability of water supply would assist NSDS processes. Synchronizing NSDS processes to follow and draw on the census data would allow the census results to be integrated into NSDS national priorities. Community development planning processes at the village level should be continuing processes, rather than a

one-off process, drawing on local strengths and run by the villages so that their results can be fed directly into NSDS. The advantage of TSDFI was that it had a less complex structure with objectives identifiable at the village level. A simple structure concentrating on island-relevant issues such as:

- Health
- Food supply, fisheries, and agriculture
- Land and marine environments, climate, and extreme events
- Infrastructure services
- Economic opportunities and employment
- Education and training
- Culture and security
- Governance and international relations

could be used as a template for village CDPs to feed directly into NSDS, with water and sanitation being covered under infrastructure, health, environment, and governance.

In the four years since their presentation in 2016, none of the priorities in CDPs had been analyzed until now. There is a pressing need for other identified priorities in the CDPs to be analyzed and incorporated into a revised TSDFII and updated ministry operational plans and reports. Without that, villagers will lose confidence in the relevance of both national and local planning processes.

We have had here the opportunity to analyze the priorities given to water and sanitation in top-down NSDSs and nation-wide, bottom-up village level Community Development Plans. The differences found here raise the question of whether NSDSs in other Pacific Island countries have the same mismatches and have to cope with similar differences between urban, rural and outer island development priorities.

**Supplementary Materials:** The following are available online at http://www.mdpi.com/2071-1050/12/22/9379/s1, Table S1: Sanitation Priorities in Community Development Plans, Tonga.

**Author Contributions:** Conceptualization, I.W., T.F. and T.K.; methodology, I.W., T.F. and T.K.; formal analysis, T.F. and I.W.; writing—original draft preparation, I.W.; writing—review and editing, I.W., T.F. and T.K.; project administration, T.K. All authors have read and agreed to the published version of the manuscript.

**Funding:** This research received no external funding.

**Acknowledgments:** This work arose as a result of a strategic planning project supported by the World Meteorology Organization under its Climate Risk and Early Warning Systems Initiative for Pacific Island Countries focusing on improved governance. WMO Pacific are thanked for that support. We are grateful to Karin Nagorcka for editorial assistance. The authors thank an unnamed reviewer whose comments improved this paper.

**Conflicts of Interest:** The authors declare no conflict of interest.

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
