# Peer review of "National Versus Local Sustainable Development Plans and Island Priorities in Sanitation: Examples from the Kingdom of Tonga"

_sustainability, doi:10.3390/su12229379_

Round 1
Reviewer 1 Report
This article is really very interesting, of a good level, it provides data that can easily be used by students and analyzes useful for the general knowledge of the phenomena discussed.
Author Response
Review 1
We thank the reviewer for generous and kind comments and for the time taken to review the paper.
Reviewer 2 Report
In Abstract, pag. 1: Please include in more clear way the aims and outcomes of your paper.
In subchp. 3.3 Changes in Sanitation Systems”, pag.12: Figure 5. The correlation coefficients associated to eq. (3) R = 0.99 and (4) R =0.96 are very high, but the number of WSR and STR values are quite reduced. Please comment on this matter.
In subchp. 3.5 „Sanitation Priorities in TSDFII”, pag. 15, rows 480 and 482: it’s not clear if the the mentioned Annex 1 is part of the TSDFII; please clarify this aspect.
In subchp. 3.6. „Sanitation Priorities in Village CDPs”, pag. 18: Figure 7. The correlation coefficients associated to eq. (5) R = 0.98 and (6) R =0.97 are very high, but the number of PSC and STR values are quite reduced.Please comment on this matter.
In subchp. 3.6. „Sanitation Priorities in Village CDPs”, pag. 19, rows 571-572: The sentence „It is noted that within error the coefficients in equations (7) and (8) are indistinguishable” is unclear; please clarify it.
In subchp. 4.4 „Sanitation in Islands reliant on Rainwater Harvesting”, pag. 22, rows 725-726: The sentence „The exception was Ongo Niua, where, in some islands, island geology means that groundwater appears absent. is very limited”, is unclear; please clarify it.
Author Response
Responses to Reviewers’ Comments
National versus Local Sustainable Development Plans and Island Priorities in Sanitation: Examples from the Kingdom of Tonga
Ian White, Tony Falkland and Taaniela Kula
Review 2
The author’s appreciate both the time taken to review our paper and for the very helpful comments which have corrected some errors in the paper and have improved the paper. We now address each of the comments.
We have now changed the abstract to:
Abstract: Sanitation, water supply and their governance remain major challenges in many Pacific Island Countries. National sustainable development strategies (NSDSs) are promoted throughout the Pacific as overarching improved governance instruments to identify priorities, plan solutions and fulfill commitments to sustainable development. Their relevance to local village-level development priorities is uncertain. In this work we compare national priorities for sanitation in NDSDs with those in village Community Development Plans (CDPs) and with metrics in Censuses from the Kingdom of Tonga. Tonga’s Strategic Development Frameworks (TSDFI 2011-2014 and TSDFII 2015-2025) were developed to focus government and its agencies on national outcomes. From 2007 to 2016, 136 villages throughout Tonga’s five Island Divisions (IDs) formulated CDPs involving separately 80% of women, youth and men in each village. It is shown that Censuses in 2006 and 2016 reveal linked improvements in water supply and sanitation systems but identify IDs with continuing challenges. It is found that sanitation and water are a national priority in TSDFI but are absent from the current TSDFII. In contrast, analysis of CDPs, published just after TSDFII, show in one ID, 53% of villages ranked sanitation as a priority and marked differences were found between IDs and between women, youth and men. CDPs’ sanitation priorities in IDs are shown to mostly correspond to sanitation and water metrics in the Censuses, but some reflect impacts of natural disasters. Explanations for differences in sanitation priorities between the national and local development plans, as well as suggestions for improving NSDS processes in island countries, are advanced.
The reviewer is quite correct. The correlation coefficients were erroneous. We have now replaced them with the correct coefficients of determination, R2 = 0.93 and R2 = 0.78 and have amended Figure 5. The statistical significance of the equations remains as given originally.
We have now clarified that Annex 1 is included in TSDFII:
Sanitation is not mentioned in any of the 29 OOs or 153 SCs and it does not appear in any of the KPIs. The word “sanitation” only appears twice in TSDFII. In Annex 1 of TSDFII, “water and sanitation management” is mentioned as one of a long list of action topics of the Small Developing Island Countries Action Agenda in the SAMOA Pathway [3].
Again, the reviewer is quite correct. We have replaced the correlation coefficients with the coefficients of determination R2 = 0.93 and R2 = 0.96. The significance level of the relationships remain as given in the manuscript.
We have now clarified this issue and have written:
It is noted that within the standard errors of the coefficients in equations (7) and (8), the two coefficients are not statistically different (p > 0.1). Combining the 2006 and 2016 data for all IDs except Ongo Niua results in the relationship:
PSC(%) = (73±9).WSR,
We have now clarified this by writing:
The exception was Ongo Niua, where, in some islands, their volcanic geology means that groundwater appears absent or is very limited.
Again the authors greatly appreciate the reviewer’s comments which have corrected and improved the paper.
